# Modifier Factors of Cystic Fibrosis Phenotypes: A Focus on Modifier Genes

**DOI:** 10.3390/ijms232214205

**Published:** 2022-11-17

**Authors:** Julie Mésinèle, Manon Ruffin, Loïc Guillot, Harriet Corvol

**Affiliations:** 1Sorbonne Université, Inserm U938, Centre de Recherche Saint-Antoine (CRSA), 75012 Paris, France; 2Inovarion, 75005 Paris, France; 3Sorbonne Université, Assistance Publique-Hôpitaux de Paris (AP-HP), Hôpital Trousseau, Service de Pneumologie Pédiatrique, 75012 Paris, France

**Keywords:** cystic fibrosis, modifiers genes, environmental factors, CFTR (Cystic Fibrosis Transmembrane Regulator) modulator therapy

## Abstract

Although cystic fibrosis (CF) is recognized as a monogenic disease, due to variants within the *CFTR* (*Cystic Fibrosis Transmembrane Regulator*) gene, an extreme clinical heterogeneity is described among people with CF (pwCF). Apart from the exocrine pancreatic status, most studies agree that there is little association between *CFTR* variants and disease phenotypes. Environmental factors have been shown to contribute to this heterogeneity, accounting for almost 50% of the variability of the lung function of pwCF. Nevertheless, pwCF with similar *CFTR* variants and sharing the same environment (such as in siblings) may have highly variable clinical manifestations not explained by *CFTR* variants, and only partly explained by environmental factors. It is recognized that genetic variants located outside the *CFTR* locus, named “modifier genes”, influence the clinical expression of the disease. This short review discusses the latest studies that have described modifier factors associated with the various CF phenotypes as well as the response to the recent CFTR modulator therapies.

## 1. Introduction

Cystic fibrosis (CF) affects about 80,000 patients worldwide. It is caused by variants in the gene coding for the chloride channel CFTR (Cystic Fibrosis Transmembrane conductance Regulator) ubiquitously expressed in epithelia [1]. *CFTR* gene variants can cause defects of synthesis or of function of the CFTR protein in epithelial cells, leading to a reduction in the permeability to chloride ions and a lack of hydration of secretions, which become viscous and sticky [2,3]. The disease affects predominately the lungs, gastrointestinal tract, pancreas, liver and biliary tract, and also the sweat glands, genital tract and osteoarticular system.

More than 2000 variants in *CFTR* have been described and assembled into six classes according to their functional consequences on the protein, with alteration of either: (I) its production; (II) its addressing to the plasma membrane; (III) its regulation (gating variant); (IV) its conduction; and (V and VI) the stability of the mRNA or of the mature protein [2,3,4]. Depending on the impact on the protein, these classes are further described as ‘minimal function variant’ (classes I, II and III) or ‘residual function variant’ (classes IV, V and VI). This classification has therapeutic implications as CFTR modulator therapies target some specific classes [2,3,4]. Exocrine pancreatic function status, whether deficient or preserved, is also associated with *CFTR* variants. Indeed, people with CF (pwCF) with pancreatic insufficiency are those carrying two minimal function variants and are more prone to suffer a severe CF phenotype, whereas pwCF with normal pancreatic function carry at least one residual function variant and have less clinical impairment [2,3,4].

Since the description of the disease, numerous observations have documented the extreme clinical heterogeneity of the disease. The identification of the *CFTR* gene and its principal variants has led to many studies aimed at establishing relationships between genotypic characteristics and clinical manifestations [5,6,7]. Apart from the exocrine pancreatic status, most studies, including those carried out in twins and siblings, agree in concluding that there is little association between *CFTR* variants and disease phenotypes [7,8,9]. Environmental factors have been shown to contribute to this heterogeneity, accounting for almost 50% of the variability of the lung function of pwCF [10]. Nevertheless, pwCF with similar *CFTR* variants and sharing the same environment (such as in siblings) may have highly variable clinical manifestations [11] not explained by *CFTR* variants, and only partly explained by environmental factors. It is recognized that genetic variants located outside the *CFTR* locus and called “modifier genes” influence the clinical expression of the disease [6,12,13]. This short review discusses the latest studies that have described modifier factors associated with the various CF phenotypes as well as the response to the recent CFTR modulator therapies.

## 2. Socio-Demographic and Environmental Modifier Factors

Several studies have highlighted the contribution of demographic and clinical factors in the variability of respiratory disease. It has been shown that women and pwCF with poor nutritional status are at greater risk of severe disease [14,15]; pwCF with CF-related diabetes (CFRD) have a more rapid decline in respiratory function [16]; CF-liver disease (CFLD) increases susceptibility to infection [17]; and the age at chronic colonization with *Pseudomonas aeruginosa* (*Pa*) decreases in the most recent cohorts [18,19]. Nevertheless, many of these factors have been questioned and remain controversial [20,21].

Regarding environmental factors, the heterogeneity of phenotypes could be modulated by passive smoking [22,23], temperature [24], outdoor [25,26,27,28] and indoor air pollution [29], seasonality [30], climate and geography [31], socioeconomic status and access to health care [32,33]. The contribution of environmental factors is nevertheless variable across the phenotypes, ranging from 40% for variability in airway obstruction to 20% for nutritional status [5].

## 3. Modifier Genes

### 3.1. Genetic Polymorphism

Modifier genes are genetic variants or polymorphisms, i.e., individual variations in the sequence of a gene, which are not pathological in themselves but can modulate disease severity [5,6]. A genetic polymorphism is determined by the coexistence of several alleles for a given gene or locus. A gene is said to be polymorphic if at least two alleles exist at a frequency greater than 1% in the population. Several types of polymorphisms can be distinguished, including minisatellite and microsatellite markers, deletions or insertions of short nucleotide sequences and single nucleotide polymorphisms (SNPs). SNPs correspond to a substitution of one purine or pyrimidine base for another one. When they are located in the coding region of a gene, they may lead to a modification of function due to the modification of the transcription of the mRNA (messenger ribonucleic acid) stability or of the protein conformation. SNPs located outside of genes have less obvious functional consequences and are generally used as markers for genetic linkage studies. SNPs are widespread, with an estimated frequency of one SNP in every 300 to 1000 base pairs [34], so more than 84.7 million SNPs are in the human genome [35]. These variations are mostly silent but are thought to be involved in the susceptibility to certain diseases, their severity, or the response to treatments.

### 3.2. A Priori and Non-A Priori Approaches

Two approaches have been applied to search for modifier genes in CF: an a priori or candidate-gene approach based on the known pathophysiology of the phenotypic trait; and a non-a priori approach carried out by whole genome analysis (GWAS for genome wide association study or WGS for whole genome sequencing) and by exome sequencing (WES for whole exome sequencing). To provide convincing statistical evidence for association and to rule out associations due to biases, these types of analyses need to be conducted in large cohorts and then replicated in independent cohorts. Thus, consortia were formed in France and North America. In France, our team has been coordinating a national study since 2006, in which nearly 5000 French pwCF are participating out of 7500 followed up in CF expert centers [36]. The merger of these two consortia has led to the creation of an international consortium that initially analyzed the GWAS of 5000 North American and 1250 French pwCF. Several regions of the genome have been shown to be associated with severity of the respiratory disease [37], risk of meconium ileus [38,39], exocrine [40] and endocrine pancreatic status [41,42], sweat chloride levels [43] and immune response [44].

### 3.3. Modifier Genes of the Lung Function

Lung function is the most studied phenotype in the search for modifier genes in CF lung disease, and multiple factors of variability have been suggested. Lung function is usually estimated by measurements of the forced expiratory volume in one second (FEV1) expressed as percent predicted of reference values (ppFEV1). It has been estimated that the lung function severity would have a heritability of more than 50% independent of CFTR [11]. The gene-candidate approach identified four main families of modifier genes of lung function based on their physiological role: genes involved in either tissue repair, host defense and inflammation, epithelial surface ion transport and mucus secretion, and response to drug therapy (detailed in the reviews of Butnariu et al. [45] and Sepahzad et al. [46]). Regarding the non-a priori approach, three (meta) GWAS were conducted, which identified seven genomic regions of interest associated with the severity of lung function (Table 1) [37,47,48].

### 3.4. Modifier Genes of Pseudomonas aeruginosa Infection

CF lung disease is caused by structural lung damage due to an exacerbated inflammation associated with recurrent bacterial infections. *Pa* is one of the most commonly isolated pathogens in the airways of pwCF. After *Pa* initial acquisition (*Pa*-IA), and despite effective eradication therapies, persistent infection may occur leading to *Pa* chronic colonization (*Pa*-CC), which is extremely difficult, if not impossible, to eradicate.

Studies conducted in twins and siblings with CF estimate that *Pa*-CC is highly heritable, up to 85% [49], with a contribution of modifier genes thought to be around 55% compared to ~23% for *CFTR* variants and ~23% for environmental factors [5]. These data attest the important role of genetic factors in the variability of the infectious phenotype of pwCF. Two non-a priori studies (WES) identified *Caveolin 2* (*CAV2*), *Dynactin subunit 4* (*DCTN4*) and *Transmembrane channel like* 6 (*TMC6*) as modifier genes of age in *Pa*-CC [50,51]. Several candidate gene studies highlighted the role of modifiers of *Pa*-IA (as *Solute carrier family 9 member 3* (*SLC9A3*) and *Solute carrier family 6 member 14* (*SLC6A14*) [52]); *Pa*-CC (as *Taste receptor 2 member 38* (*TAS2R38*) [53] and *Interleukin-10* (*IL10*) [54]); and both *Pa*-IA and *Pa*-CC (as *Tumor necrosis factor* (*TNF*) [55] and *Mannose-binding lectin 2* (*MBL2*) [56,57]).

We recently analyzed a large cohort of 1231 French children with CF to identify risk factors and genetic modifiers for *Pa*-IA, *Pa*-CC and progression from *Pa*-IA to *Pa*-CC [19].We found that *Pa* infection occurred early in childhood, with the median age of patients being 5.1 years at *Pa*-IA. One-quarter of the pwCF were chronically colonized with *Pa* by the age of 14.7 years, within 6.3 years after *Pa*-IA. We highlighted that *Pa*-CC occurs later in life for pwCF in the most recent birth cohorts, which may reflect improvements in the standardization of therapeutic care, especially since the establishment of specialized CF care centers. Interestingly, we observed that CFRD and CFLD were risk factors for *Pa*, while gender, *CFTR* variants and CF center size were not. Moreover, we confirmed that genetic variants of *TNF*, *DCTN4*, *SLC9A3* and *CAV2* were associated with *Pa*.

### 3.5. Modifier Genes of Meconium Ileus

Around 15% of pwCF are born with an intestinal obstruction at birth named meconium ileus (MI). A first GWAS identified several genes associated with MI susceptibility, especially *SLC6A14*, *SLC26A9* (*Solute carrier family 26 member 9*) and *SLC9A3* [38]. These genes belong to the SLC (solute carriers) family, which encodes membrane transport proteins. The different members of the SLC family transport a large panel of substances including charged and uncharged organic molecules as well as inorganic ions. A more recent meta-GWAS for MI susceptibility confirmed the previously identified *SLC26A9* and *SLC6A14* and found two new loci, *ATP12A* (*ATPase H+/K+ transporting non-gastric alpha2 subunit*) and *PRSS1* (*Serine protease 1*) [39].

### 3.6. Modifier Genes of CF-Related Diabetes

CFRD is a frequent complication of CF. Its prevalence increases with age, affecting more than 90% of pwCF carrying two minimal function *CFTR* variants by their sixth decade [41]. Age at onset varies considerably (e.g., from 10 to 50 years old) and has been shown to be highly heritable but independent of *CFTR* genotype, suggesting an important role for modifiers genes.

An initial gene-candidate study highlighted that the *TCF7L2* (*Transcription factor 7-like 2*) gene, known to be associated with type 2 diabetes in the general population, was also associated with CFRD [58]. This association was confirmed in an early GWAS, which also identified the *SLC26A9* gene and genomic regions next to the *CDKAL1* (*Cyclin dependent kinase 5 regulatory associated protein 1-like 1*), *CDKN2A/B* (*Cyclin dependent kinase inhibitor 2A/B*) and *IGF2BP2* (*Insulin-like growth factor-binding protein 2*) genes [59]. More recently, the *SLC26A9* gene has been confirmed in a candidate gene study [60] as well as a meta-GWAS [41]. The later meta-GWAS also replicated the involvement of the *TCF7L2* gene and identified the *PTMA* (*Prothymosin alpha*) gene [41].

Innovatively, using genetic and clinical measures available at birth, a CFRD prediction model has been recently constructed [42]. This model found that the strongest CFRD predictors included sex, CFTR severity score and several genetic variants. A web-based application was developed to provide practitioners with patient-specific CFRD risk to guide monitoring and treatment.

### 3.7. Modifier Genes of CF-Liver Disease

CFLD includes various hepatobiliary abnormalities. Focal biliary cirrhosis is the most clinically relevant in CFLD, and extension of the initially focal fibrogenic process may indeed lead to multilobular biliary cirrhosis with subsequent portal hypertension and related complications [61,62,63]. Modifier genes of CFLD have so far only been investigated by gene-candidate approaches. Studies have explored the potential role of *SERPINA1* (*Alpha-1 Antitrypsine)* [64], *ACE* (*Angiotensin I-converting enzyme*) [65], *GSTP1* (*Glutathione S-transferases P1*) [66], *MBL2* [67,68], *TGFβ1* (*Transforming growth factor beta 1*) [65] and *ABCB4* (*ATP Binding cassette subfamily B member 4)* [68]. The association between the *SERPINA1* gene and CFLD severity was the only one to be successfully replicated, showing a highly increased risk for pwCF carrying the Z allele [69,70]. To date, no GWAS analysis has been conducted. Nevertheless, one study has highlighted the probable association between CFLD and six new genes using a system biology approach to investigate the functional relationship between the previously studied genes [71]. This particular methodology combines database and literature mining, gene expression study and network analysis as well as pathway enrichment analysis and protein–protein interactions.

## 4. Modifier Factors of CFTR Modulators Response

The management of CF, which was essentially symptomatic until recently, has led to a marked improvement in survival, which now exceeds 50 years [36]. These *symptomatic* treatments target the side effects of CFTR protein dysfunction. For respiratory disease, for example, treatments consist of improving secretion drainage, mucus hydration and/or mucociliary clearance, as well as treating recurrent infections or exacerbated inflammation [72]. In the last decade, considerable efforts have led to the development of therapies that target the CFTR proteins named CFTR modulators [73]. Interpatient variability in the response to these modulators has been shown to be partly explained by modifier genes.

### 4.1. Modifier Genes and Response to Ivacaftor

Since 2012, pwCF carrying various *CFTR* gating variants could be treated with ivacaftor, a potentiator therapy that increases the open probability of CFTR-channels. Its efficacy was initially demonstrated in pwCF carrying at least one G551D variant and was later extended to other less common gating variants [74]. Ivacaftor was shown to allow the recovery of 30 to 40% of ‘normal’ CFTR function and is associated with significant clinical benefits, such as gain of lung function and reduced number of pulmonary exacerbations and hospitalizations [75]. Significant inter-patient variability in response to ivacaftor is observed, particularly with regard to lung response. Initiation of ivacaftor treatment is indeed associated with a mean ppFEV1 improvement of 10.4% at 24 weeks of treatment with a variability of ±15.5% between pwCF [76]. In French pwCF, we have shown that the variability of the respiratory response to ivacaftor was associated with the *SLC26A9* gene [77], confirming the results observed in Canadians [78]. However, these results were recently controverted in pwCF from the US [79].

### 4.2. Modifier Genes and Response to Lumacaftor-Ivacaftor

Lumacaftor, a CFTR corrector that improves the processing and trafficking of the Phe508del-CFTR protein, was combined with ivacaftor to treat pwCF homozygous for the Phe508del-CFTR variant. A phase 2 trial of this combined therapy, lumacaftor–ivacaftor (LUMA–IVA), demonstrated improvements in lung function, measured by an increase of the ppFEV1, and nutritional status, evaluated by the body mass index (BMI) [80]. Since the marketing authorizations granted in 2015, the clinical benefits of LUMA–IVA have been questioned, particularly because of its high cost [81]. In addition, phase 3 trials and real-life studies over the first 2 years of treatment demonstrated highly heterogeneous responses in lung function and nutritional status [82,83,84,85,86,87,88]. These studies also highlighted the inter-individual variability of the airway response and the limited tolerance to treatment, with high discontinuation rates ranging from 17.2% to 28.9%. Predisposing factors for interruption and for the response variability were shown to be baseline lung function, age at treatment initiation and gender [82,84,85,86,87,88].

Our team has very recently been able to identify factors, including modifier genes, influencing the respiratory and nutritional response to LUMA–IVA [89]. Among the 4981 pwCF of our gene-modifiers cohort, 878 were eligible for analysis of the respiratory and nutritional response to LUMA–IVA, and 765 were included. We evaluated the respiratory and nutritional responses after 1 to 2 years of LUMA–IVA and then looked for clinical and genetic factors involved in their variability. We analyzed *SLC* variants that had been implicated in the variability of response to ivacaftor [77,78] and/or in the severity of several CF phenotypes [37,38,52,90,91,92]. Gains in lung function and nutritional status were observed after 6 months of treatment (on average 2.11 ± 7.81% for ppFEV1 and 0.44 ± 0.77 kg/m^2^ for BMI) and sustained over the 2 years. We observed that the more severe pwCF gained the most in lung function and nutritional status. While females started with a nutritional status more impaired than males, they had a larger response and regained BMI Z-score values similar to men after 2 years of treatment. We observed no association between *SLC* variants and the respiratory function response to LUMA–IVA, but *SLC6A14* rs12839137 was associated with the nutritional response.

### 4.3. Modifier Genes and Response to Tezacaftor-Ivacaftor and Elexacaftor-Tezacaftor-Ivacaftor

Therapies combining the modulators tezacaftor–ivacaftor (TEZ–IVA) and elexacaftor–tezacaftor–ivacaftor (ETI) have been marketed more recently. Huge improvements in lung function and nutritional status have been shown in treated patients [93,94,95,96]. So far, no phenotype/genotype association study has investigated the role of modifier genes in the variability of the response to these therapies. Nevertheless, two studies suggest *SLC34A2* (*Solute carrier family 34 member 2*) and *SLC26A9* as potential modifiers [97,98].

## 5. Conclusions and Perspectives

Advances in the understanding of CFTR functions have led to amazing therapeutic progresses. Nevertheless, the observed phenotypic variability highlights the need to better understand the differences between subgroups of pwCF in order to identify prognostic indicators for optimal patient management. More specifically, the identification of modifier genes and a better knowledge of their functions in CF could allow the development of personalized, predictive, preventive and participatory medicine [4]. Several modifier genes of different CF phenotypes have been identified so far, as detailed in this short review (Figure 1).

However, many studies have been carried out using candidate gene studies, which have not been replicated and for which the functionality of the identified genes or variants has not been deeply investigated. The challenge for the future is, as it was initiated for lung function, CFRD and MI, to perform studies without preconceptions to identify modifier genes and variants involved in CFLD infections, as well as validate these modifiers at the functional level. This is necessary to move on to the next step which will consist of establishing polygenic risk scores (PRS) that could predict the degree of severity of the various CF disorders, as realized with CFRD [42], and thus allow targeted and individualized management of pwCF.

To conclude, genomic knowledge will enable individualized medicine through the development of new therapies or by adapting existing ones more specifically. Furthermore, it will facilitate predictive and preventive medicine through the evaluation of the risks of developing complications, which is equivalent to a proactive approach to patient management and oriented towards the preservation of health capital.

## Figures and Tables

**Figure 1 ijms-23-14205-f001:**
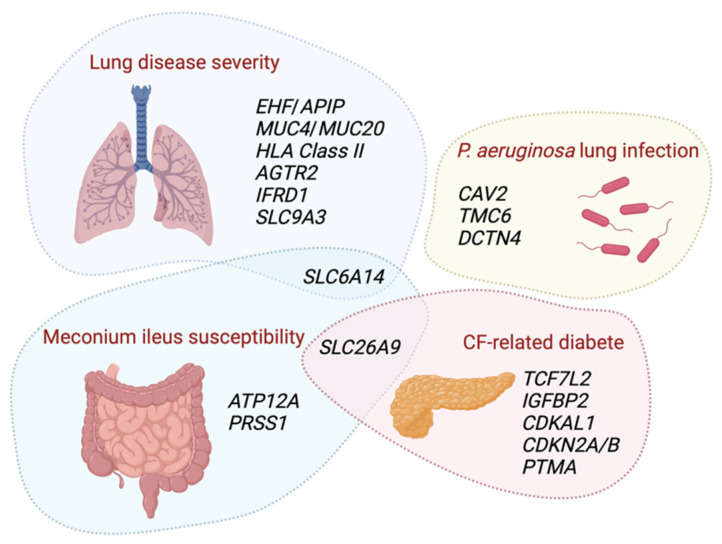
Genetic modifiers of cystic fibrosis (CF) phenotypic traits. Schematic representation of genes identified by genome wide association studies (GWAS) and whole exome sequencing (WES) as associated with the heterogeneity of CF complications (lung disease severity, *Pseudomonas. aeruginosa* lung infection, meconium ileus susceptibility and CF-related diabetes). Abbreviations (alphabetic order): *angiotensin II receptor type 2 (AGTR2), Apaf-1 interacting protein (APIP), ATPase H^+^/K^+^ transporting non-gastric alpha2 subunit (ATP12A), Caveolin 2 (CAV2), class II major histocompatibility complex (HLA Class II), cyclin dependent kinase 5 regulatory associated protein 1-like 1 (CDKAL1), cyclin dependent kinase inhibitor 2A/B (CDKN2A/B), Dynactin subunit 4 (DCTN4), Ets Homologous Factor (EHF), Interferon Related Developmental Regulator 1 (IFRD1), insulin-like growth factor-binding protein 2 (IGFBP2), mucin 4 (MUC4), mucin 20 (MUC20), prothymosin alpha (PTMA), serine protease 1 (PRSS1), solute carrier family 26 member 9 (SLC26A9), solute carrier family 6 member 14 (SLC6A14), solute carrier family 9 member A3 (SLC9A3), transcription factor 7-like 2 (TCF7L2) and Transmembrane channel like 6 (TMC6).* Created with BioRender.com.

**Table 1 ijms-23-14205-t001:** Modifier genes associated with the inter-individual variability of lung function in people with cystic fibrosis and identified by genome wide association studies (sorted alphabetically).

Modifier Gene or Locus	Physiological Role	Ref.
*Angiotensin II receptor type 2 (AGTR2)/Solute carrier family 6 member A14 (SLC6A14)* *Locus chrXq22-q23*	AGTR2 encodes the angiotensin, which is part of the renin-angiotensin system, a major control system for blood pressure and fluid balance. SLC6A14 imports and concentrates neutral amino acids and the two cationic acids lysine and arginine into the cytoplasm of different cell types.	[37]
*ETS homologous factor (EHF)/APAF1 interacting protein (APIP)* *Locus chr11p12-p13*	EHF transcription factor modulates epithelial tight junctions and wound repair. APIP is a methionine salvage pathway enzyme associated with apoptosis and systemic inflammatory responses.	[37] [47]
*Human leukocyte antigen (HLA) Class II* *Locus chr6p21.3*	HLA plays a central role in immunity by presenting peptides derived from extracellular proteins.	[37]
*Interferon related developmental regulator 1 (IFRD1)* *Locus chr11p12-p13*	IFRD1 is as a transcriptional co-activator/repressor regulating cellular growth and differentiation during tissue development and regeneration.	[48]
*Locus chr 20q13.2*	The five genes located in this locus are expressed in respiratory epithelial cells. Among these genes, the melanocortin 3 receptor (MC3R) is implicated in weight maintenance and regulation of energy balance. Some of these genes encode proteins involved in cell adhesion, migration and phagocytosis of bacterial pathogens.	[47]
*Mucin 4 (MUC4)/Mucin 20 (MUC20)* *Locus chr3q29*	Mucins are highly glycosylated mucus proteins.	[37]
*Solute carrier family 9 member A3 (SLC9A3)*	SLC9A3 is a sodium–hydrogen exchanger.	[37]

## Data Availability

Not applicable.

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
