# Peer review of "Modifier Factors of Cystic Fibrosis Phenotypes: A Focus on Modifier Genes"

_ijms, 2022, doi:10.3390/ijms232214205_

Round 1
Reviewer 1 Report
The Review by Mésinèle et al summarizes shortly and precisely our current understanding of modifier factors in relation to the wide variations of cystic fibrosis associated symptoms in the patients. The authors delved deep into wide-ranging literature and they provide a thorough compendium of citations with short comments on the respective findings. Although the paper represents a solid piece of work, I have some points that should be properly addressed before considered acceptable.
1. In table 1 (page 4, last paragraph) the SLC9A3 it is stated that the protein uses a Na+ gradient to eject amino acids from the cells. Instead, the NHE 3 antiporter imports one Na+ into the cytosol of a cell as it ejects one hydrogen ion (or proton) from the cell into extracellular space. The resulting proton gradient might be used for a tertiary active transport system to export amino acids. This fact must be corrected.
2. In an extensive paragraph the authors describe the interactions of modifier factors and CFTR modulators using the examples of Ivacaftor and Lumacaftor. However, in the treatment of CF patients we see the most promising effects using Kaftrio (Trikafta in the US) containing additionally the modulators Tezacaftor and Elaxacaftor. To my opinion it is mandatory for a state-of-the-art review in this journal to include these most recent modulators and their known interactions with modifier factors instead of simply ignoring them.
Some minor issues:
3. The authors state that some scientists contributed equally but do not identify them
4. As the whole paper is quite short is should be termed Mini Review
5. The vast majority of all journals use the term ‘CF patients’, therefore it’s not necessary to introduce another confusing abbreviation such as ‘pwCF’. Please change
Author Response
We thank reviewer 1 for his evaluation and criticism. Our responses are detailed below point by point.
- In table 1 (page 4, last paragraph) the SLC9A3 it is stated that the protein uses a Na+ gradient to eject amino acids from the cells. Instead, the NHE 3 antiporter imports one Na+ into the cytosol of a cell as it ejects one hydrogen ion (or proton) from the cell into extracellular space. The resulting proton gradient might be used for a tertiary active transport system to export amino acids. This fact must be corrected.
We thank the reviewer for his comment and have corrected the Table 1. We have replaced: “SLC9A3 is an epithelial brush border Na/H exchanger that uses an inward sodium ion gradient to eject acids from the cell.” by: “SLC9A3 is a sodium-hydrogen exchanger.”
- In an extensive paragraph the authors describe the interactions of modifier factors and CFTR modulators using the examples of Ivacaftor and Lumacaftor. However, in the treatment of CF patients we see the most promising effects using Kaftrio (Trikafta in the US) containing additionally the modulators Tezacaftor and Elaxacaftor. To my opinion it is mandatory for a state-of-the-art review in this journal to include these most recent modulators and their known interactions with modifier factors instead of simply ignoring them.
We thank the reviewer for this comment. There are indeed limited studies evaluating modifiers of the response to these recent therapies, i.e. tezacaftor-ivacaftor and elexacaftor-tezacaftor-ivacaftor. As suggested we have added a paragraph “4.3. Modifier genes and response to tezacaftor-ivacaftor and elexacaftor-tezacaftor-ivacaftor” (page 6 lines 241-247): “Therapies combining the modulators tezacaftor-ivacaftor (TEZ-IVA) and elexacaftor-tezacaftor-ivacaftor (ETI) have been marketed more recently. Huge improvements in lung function and nutritional status have been shown in treated patients (1-4). So far, no phenotype/genotype association study has investigated the role of modifier genes in the variability of the response to these therapies. Nevertheless, functional genetic studies have identified SLC34A2 and SLC26A9 as potential modifiers (5, 6).”
Some minor issues:
- The authors state that some scientists contributed equally but do not identify them
We apologize for this error and have deleted this statement.
- As the whole paper is quite short is should be termed Mini Review
There is no size recommendations in the author guidelines of IJMS, with no requirement to specify whether it is a mini review or not. As such, we did not change the review term but to follow the instruction of the reviewer, we added the ‘short’ in front of ‘review’:
- In the abstract page 1 line 19 : “This short review discusses the latest studies that have described modifier factors associated with the various CF phenotypes as well as with the response to the recent CFTR modulator therapies.”
- Page 2 line 57: “This short review discusses the latest studies that have described modifier factors associated with the various CF phenotypes as well as with the response to the recent CFTR modulator therapies.”
- The vast majority of all journals use the term ‘CF patients’, therefore it’s not necessary to introduce another confusing abbreviation such as ‘pwCF’. Please change
Since several years, the European and North-American CF societies (ECFS and NACFC respectively) recommend the use of ‘people with CF’ with the abbreviation ‘pwCF’ instead of ‘patients with CF’. In order to remain consistent with this terminology, especially for patients, we prefer not to change this abbreviation.
Reviewer 2 Report
Although the manuscript is interesting, reviewing the modifier genes in the genetic disease cystic fibrosis, several important points need to be addressed to improve the manuscript and make it more instructive and complete for the reader.
My major comments are:
1. The general comment concerning that manuscript is that the recent review by Butnariu et al 2021 (J Clin Med) is much more complete and informative expecially with a more complet list of modifier genes and a whole description of non-genetic factors. Given the contribution of the CGM (CF gene Modifier consortium) with USA, Canada and France, it is surprising that authors refers too much to their own contribution.
2. The table lists a series of modifiers without any hierarchy or logic. A number of genes are missing, especially it lacks SLC26A9, yet often cited in the text. I also would suggest to add SERPINA1 and also T2R38 (see for example Castaldo et al 2020 and Dalesio et al 2020). The excellent paper by Lam et al 2020 (from the Cutting's lab on SLC26A9 should have been quoted as well)
3. The socio-demographic and environemental factors are minimally described (only 13 lines description)
Other comments are:
4. The title of the manuscript should rather specify genetic and non-genetic modifier factors.
5. Line 42: what do you mean by "classic CF penotype" when it is emphasized the clinical heterogeneity of the disease two lines below?
6. Line 198: the effect of ivacaftor as a potentiator is to increase the open probability of CFTR channels and not the probability of channel opening
7. Line 209: "Table 1" refers to what exactly in the table?
Author Response
We thank reviewer 2 for his evaluation and criticism. Our responses are detailed below point by point.
My major comments are:
- The general comment concerning that manuscript is that the recent review by Butnariu et al 2021 (J Clin Med) is much more complete and informative especially with a more complete list of modifier genes and a whole description of non-genetic factors. Given the contribution of the CGM (CF gene Modifier consortium) with USA, Canada and France, it is surprising that authors refers too much to their own contribution.
We agree with the reviewer that the review by Butnariu et al. is well detailed, especially describing several genes identified by candidate-gene approaches. Thus, we now cite this paper as a reference for the genes that have been identified through candidate gene studies (page 3, lines 114-118): “The gene-candidate approach identified four main families of modifier genes of lung function based on their physiological role: genes involved in either tissue repair, host defense and inflammation, epithelial surface ion transport and mucus secretion, and response to drug therapy (detailed in the reviews of Butnariu et al. (7) and Sepahzad et al. (8))”.
Here, when available, we chose to mainly focus on the results of non-a priori studies (GWAS or WES) conducted in large cohorts. Our team is the French part of the CGM (CF gene Modifier consortium) with USA and Canada. This international consortium has indeed allowed us to conduct large GWAS and to identify several genomic regions associated with severity of the respiratory disease (9), risk of meconium ileus (10, 11), exocrine (12) and endocrine pancreatic status (13, 14), sweat chloride levels (15) and immune response (16). We have described this consortium and our contribution in the paragraph 3.2 "A priori and non-apriori approaches" (replacing “International CF modifier consortium”) (pages 2-3, lines 92-107).
- The table lists a series of modifiers without any hierarchy or logic. A number of genes are missing, especially it lacks SLC26A9, yet often cited in the text. I also would suggest to add SERPINA1 and also T2R38 (see for example Castaldo et al 2020 and Dalesio et al 2020). The excellent paper by Lam et al 2020 (from the Cutting's lab on SLC26A9 should have been quoted as well).
We thank the reviewer for his comment. The table being confusing, we completely changed it. It now reports only genes associated with lung function (and no more with infection) and that were identified only by non-a priori approaches (GWAS only for lung function). The genes are sorted alphabetically (which is now indicated in the title of the table).
As such, and also in agreement with the previsous comment, we changed the text of the 3.3 paragraph “Modifier genes of the lung function”, which now reads (page 3, lines 114-118): “The gene-candidate approach identified four main families of modifier genes of lung function based on their physiological role: genes involved in either tissue repair, host defense and inflammation, epithelial surface ion transport and mucus secretion, and response to drug therapy (detailed in the reviews of Butnariu et al.(7) and Sepahzad et al. (8)). Regarding the non-a priori approach, three (meta)GWAS were conducted, which identified 7 genomic regions of interest associated with the severity of lung function (Table 1) (9, 17, 18).”
We also modified the text of the 3.4 paragraph “Modifier genes of Pseudomonas aeruginosa infection”, which now reads (page 4, lines 134-137): “Two non-a priori studies (WES) identified CAV2, DCTN4 and TMC6 as modifier genes of age at Pa-CC (19, 20). Several candidate gene studies highlighted the role of modifiers of Pa-IA (as SLC9A3 and SLC6A14 (21)); Pa-CC (as TAS2R38 (22) and IL10 (23)); and both Pa-IA and Pa-CC (as TNF (24) and MBL2 (25, 26)).”
As suggested by the reviewer, we have added that the T2R38 gene was identified as possible modifier of P. aeruginosa infection with the reference of Castaldo et al. (22) (page 4, line 136). However, we did not add the reference of Dalesio et al. as it refers to sinus disease, a complication not discussed in this review.
Regarding SLC26A9, no GWAS study has shown its involvement in lung disease severity (9, 17, 18). Candidate gene studies also point in this direction: in the Canadian cohort (27)(1,759 pwCF homozygous for Phe508del), in our French cohort (28) (3,418 pancreatic insufficient pwCF), and in the US cohort (29)(272 pwCF with at least one G551D mutation). In a Brazilian cohort, SLC26A9 was suggested as associated with lung function, but the result was no longer statistically significant after P-Value correction (30). As such, SLC26A9 is not listed in Table 1. Nevertheless, we added the reference of Lam et al., where SLC26A9 is shown to be associated with CF-related diabetes onset in the 3.5 paragraph “Modifier genes of CF-related diabetes” (page 4, lines 168- 169) and modified the text accordingly: “More recently, the SLC26A9 gene has been confirmed in a candidate gene study (31) […].”
The SERPINA1 gene has indeed been identified by candidate genes studies with lung disease and Pa infection (32-35). As the table has been modified to report only genes associated with lung function identified by non-a priori approaches, SERPINA1 is not listed. Nevertheless, it is reported in the review by Butnariu et al. that we now refer to for modifiers identified though candidate gene studies (please see response to the previous comment).
- The socio-demographic and environmental factors are minimally described (only 13 lines description)
This short review focused on modifier genes of cystic fibrosis. The socio-demographic and environmental factors have been mentioned briefly because they could not be ignored in the background of inter-individual variability. To make this clearer to the reader, we now mentioned it in the title: “Modifier Factors of Cystic Fibrosis Phenotypes: Focus on modifier genes”
Other comments are:
- The title of the manuscript should rather specify genetic and non-genetic modifier factors.
As mentioned in the previous comment, we changed the title of the manuscript which now reads: “Modifier Factors of Cystic Fibrosis Phenotypes: Focus on modifier genes”
- Line 42: what do you mean by "classic CF phenotype" when it is emphasized the clinical heterogeneity of the disease two lines below?
We thank the reviewer for this comment that highlights a lack of accuracy. The sentence (page 1 lines 41 to 44) has been replaced by: “Indeed, people with CF (pwCF) with pancreatic insufficiency are those carrying 2 minimal function variants and are more prone to suffer a severe CF phenotype, whereas pwCF with normal pancreatic function carry at least one residual function variant and have less clinical impairment (36-38).”
- Line 198: the effect of ivacaftor as a potentiator is to increase the open probability of CFTR channels and not the probability of channel opening
We thank the reviewer for this comment and have corrected the text (page 6 lines 200) which now reads:“Since 2012, pwCF carrying various CFTR gating variants can be treated with ivacaftor, a potentiator therapy that increases the open probability of CFTR-channels.”
- Line 209: "Table 1" refers to what exactly in the table?
We have deleted the reference to Table 1 in the paragraph.
REFERENCES
- Rowe SM, Daines C, Ringshausen FC, Kerem E, Wilson J, Tullis E, et al. Tezacaftor-Ivacaftor in Residual-Function Heterozygotes with Cystic Fibrosis. N Engl J Med. 2017;377(21):2024-35.
- Taylor-Cousar JL, Munck A, McKone EF, van der Ent CK, Moeller A, Simard C, et al. Tezacaftor-Ivacaftor in Patients with Cystic Fibrosis Homozygous for Phe508del. N Engl J Med. 2017;377(21):2013-23.
- Middleton PG, Mall MA, Drevinek P, Lands LC, McKone EF, Polineni D, et al. Elexacaftor-Tezacaftor-Ivacaftor for Cystic Fibrosis with a Single Phe508del Allele. N Engl J Med. 2019;381(19):1809-19.
- Burgel PR, Durieu I, Chiron R, Ramel S, Danner-Boucher I, Prevotat A, et al. Rapid Improvement after Starting Elexacaftor-Tezacaftor-Ivacaftor in Patients with Cystic Fibrosis and Advanced Pulmonary Disease. Am J Respir Crit Care Med. 2021;204(1):64-73.
- Saint-Criq V, Wang Y, Delpiano L, Lin J, Sheppard DN, Gray MA. Extracellular phosphate enhances the function of F508del-CFTR rescued by CFTR correctors. J Cyst Fibros. 2021;20(5):843-50.
- Pinto MC, Quaresma MC, Silva IAL, Railean V, Ramalho SS, Amaral MD. Synergy in Cystic Fibrosis Therapies: Targeting SLC26A9. Int J Mol Sci. 2021;22(23).
- Butnariu LI, Tarca E, Cojocaru E, Rusu C, Moisa SM, Leon Constantin MM, et al. Genetic Modifying Factors of Cystic Fibrosis Phenotype: A Challenge for Modern Medicine. J Clin Med. 2021;10(24).
- Sepahzad A, Morris-Rosendahl DJ, Davies JC. Cystic Fibrosis Lung Disease Modifiers and Their Relevance in the New Era of Precision Medicine. Genes (Basel). 2021;12(4).
- Corvol H, Blackman SM, Boelle PY, Gallins PJ, Pace RG, Stonebraker JR, et al. Genome-wide association meta-analysis identifies five modifier loci of lung disease severity in cystic fibrosis. Nat Commun. 2015;6:8382.
- Sun L, Rommens JM, Corvol H, Li W, Li X, Chiang TA, et al. Multiple apical plasma membrane constituents are associated with susceptibility to meconium ileus in individuals with cystic fibrosis. Nat Genet. 2012;44(5):562-9.
- Gong J, Wang F, Xiao B, Panjwani N, Lin F, Keenan K, et al. Genetic association and transcriptome integration identify contributing genes and tissues at cystic fibrosis modifier loci. PLoS genetics. 2019;15(2):e1008007.
- Miller MR, Soave D, Li W, Gong J, Pace RG, Boelle PY, et al. Variants in Solute Carrier SLC26A9 Modify Prenatal Exocrine Pancreatic Damage in Cystic Fibrosis. J Pediatr. 2015;166(5):1152-7 e6.
- Aksit MA, Pace RG, Vecchio-Pagan B, Ling H, Rommens JM, Boelle PY, et al. Genetic Modifiers of Cystic Fibrosis-Related Diabetes Have Extensive Overlap With Type 2 Diabetes and Related Traits. J Clin Endocrinol Metab. 2020;105(5).
- Lin YC, Keenan K, Gong J, Panjwani N, Avolio J, Lin F, et al. Cystic fibrosis-related diabetes onset can be predicted using biomarkers measured at birth. Genetics in medicine : official journal of the American College of Medical Genetics. 2021;23(5):927-33.
- Collaco JM, Blackman SM, Raraigh KS, Corvol H, Rommens JM, Pace RG, et al. Sources of Variation in Sweat Chloride Measurements in Cystic Fibrosis. Am J Respir Crit Care Med. 2016.
- Polineni D, Dang H, Gallins PJ, Jones LC, Pace RG, Stonebraker JR, et al. Airway Mucosal Host Defense Is Key to Genomic Regulation of Cystic Fibrosis Lung Disease Severity. Am J Respir Crit Care Med. 2018;197(1):79-93.
- Wright FA, Strug LJ, Doshi VK, Commander CW, Blackman SM, Sun L, et al. Genome-wide association and linkage identify modifier loci of lung disease severity in cystic fibrosis at 11p13 and 20q13.2. Nat Genet. 2011;43(6):539-46.
- Gu Y, Harley IT, Henderson LB, Aronow BJ, Vietor I, Huber LA, et al. Identification of IFRD1 as a modifier gene for cystic fibrosis lung disease. Nature. 2009;458(7241):1039-42.
- Emond MJ, Louie T, Emerson J, Chong JX, Mathias RA, Knowles MR, et al. Exome Sequencing of Phenotypic Extremes Identifies CAV2 and TMC6 as Interacting Modifiers of Chronic Pseudomonas aeruginosa Infection in Cystic Fibrosis. PLoS genetics. 2015;11(6):e1005273.
- Emond MJ, Louie T, Emerson J, Zhao W, Mathias RA, Knowles MR, et al. Exome sequencing of extreme phenotypes identifies DCTN4 as a modifier of chronic Pseudomonas aeruginosa infection in cystic fibrosis. Nat Genet. 2012;44(8):886-9.
- Li W, Soave D, Miller MR, Keenan K, Lin F, Gong J, et al. Unraveling the complex genetic model for cystic fibrosis: pleiotropic effects of modifier genes on early cystic fibrosis-related morbidities. Hum Genet. 2014;133(2):151-61.
- Castaldo A, Cernera G, Iacotucci P, Cimbalo C, Gelzo M, Comegna M, et al. TAS2R38 is a novel modifier gene in patients with cystic fibrosis. Sci Rep. 2020;10(1):5806.
- Tesse R, Cardinale F, Santostasi T, Polizzi A, Mappa L, Manca A, et al. Association of interleukin-10 gene haplotypes with Pseudomonas aeruginosa airway colonization in cystic fibrosis. J Cyst Fibros. 2008;7(4):329-32.
- Coutinho CA, Marson FA, Marcelino AR, Bonadia LC, Carlin MP, Ribeiro AF, et al. TNF-alpha polymorphisms as a potential modifier gene in the cystic fibrosis. Int J Mol Epidemiol Genet. 2014;5(2):87-99.
- Trevisiol C, Boniotto M, Giglio L, Poli F, Morgutti M, Crovella S. MBL2 polymorphisms screening in a regional Italian CF Center. J Cyst Fibros. 2005;4(3):189-91.
- McDougal KE, Green DM, Vanscoy LL, Fallin MD, Grow M, Cheng S, et al. Use of a modeling framework to evaluate the effect of a modifier gene (MBL2) on variation in cystic fibrosis. Eur J Hum Genet. 2010;18(6):680-4.
- Strug LJ, Gonska T, He G, Keenan K, Ip W, Boelle PY, et al. Cystic fibrosis gene modifier SLC26A9 modulates airway response to CFTR-directed therapeutics. Hum Mol Genet. 2016;25(20):4590-600.
- Corvol H, Mésinèle J, Douksieh I-H, Strug LJ, Boëlle P-Y, Guillot L. SLC26A9 Gene Is Associated With Lung Function Response to Ivacaftor in Patients With Cystic Fibrosis. Frontiers in Pharmacology. 2018;9(828).
- Eastman AC, Pace RG, Dang H, Aksit MA, Vecchio-Pagan B, Lam AN, et al. SLC26A9 SNP rs7512462 is not associated with lung disease severity or lung function response to ivacaftor in cystic fibrosis patients with G551D-CFTR. J Cyst Fibros. 2021.
- Pereira SV, Ribeiro JD, Bertuzzo CS, Marson FAL. Association of clinical severity of cystic fibrosis with variants in the SLC gene family (SLC6A14, SLC26A9, SLC11A1 and SLC9A3). Gene. 2017;629:117-26.
- Lam AN, Aksit MA, Vecchio-Pagan B, Shelton CA, Osorio DL, Anzmann AF, et al. Increased expression of anion transporter SLC26A9 delays diabetes onset in cystic fibrosis. J Clin Invest. 2020;130(1):272-86.
- Courtney JM, Plant BJ, Morgan K, Rendall J, Gallagher C, Ennis M, et al. Association of improved pulmonary phenotype in Irish cystic fibrosis patients with a 3' enhancer polymorphism in alpha-1-antitrypsin. Pediatr Pulmonol. 2006;41(6):584-91.
- Mahadeva R, Stewart S, Bilton D, Lomas DA. Alpha-1 antitrypsin deficiency alleles and severe cystic fibrosis lung disease. Thorax. 1998;53(12):1022-4.
- Henry MT, Cave S, Rendall J, O'Connor CM, Morgan K, FitzGerald MX, et al. An alpha1-antitrypsin enhancer polymorphism is a genetic modifier of pulmonary outcome in cystic fibrosis. Eur J Hum Genet. 2001;9(4):273-8.
- Doring G, Krogh-Johansen H, Weidinger S, Hoiby N. Allotypes of alpha 1-antitrypsin in patients with cystic fibrosis, homozygous and heterozygous for deltaF508. Pediatr Pulmonol. 1994;18(1):3-7.
- Corvol H, Taytard J, Tabary O, Le Rouzic P, Guillot L, Clement A. [Challenges of personalized medicine for cystic fibrosis]. Archives de pediatrie : organe officiel de la Societe francaise de pediatrie. 2015.
- Corvol H, Thompson KE, Tabary O, le Rouzic P, Guillot L. Translating the genetics of cystic fibrosis to personalized medicine. Transl Res. 2016;168:40-9.
- Elborn JS. Cystic fibrosis. Lancet. 2016;388(10059):2519-31.
Round 2
Reviewer 1 Report
Altough I still do not agree with the authors use of the term 'pwCF' all of my former suggestions were addressed properly, therefore the manuscript might be acceptable
Reviewer 2 Report
Thank you for the responses regarding the original manuscript